# The Impact of Grassroots Forestry Institutions on Forest Carbon Sequestration: Evidence from China's Collective Forests

**Shuqiang Li, Li Gao, Khan Hassan Saif and Hua Li ***

School of Economics and Management, Northwest A & F University, Xianyang 712100, China; lishuqiang@nwafu.edu.cn (S.L.)
* Correspondence: lihua7485@nwafu.edu.cn

**Abstract:** Many countries have established grassroots forestry institutions to manage and protect small-scale forestry resources and provide technology and services to private foresters. Since the inception of township forestry workstations (TFWs) in China almost 70 years ago, TFW has supported resource protection and forest property reform. In this paper, we employ fixed effect models to test the effects of TFW on collective forest carbon density and provide evidence for improving the quality of collective forests. Our results demonstrate that TFWs in China improve the carbon density of collective forests by performing forestry management and service functions. However, significant differences in TFWs exist under different management systems, and the dual leadership township forestry workstation (D_TWF) is more effective in increasing the carbon density of collective forests. The management system's heterogeneity directly affects its performance, with D_TWF performing better management functions and the single leadership township forestry workstation (S_TWF) performing better service functions. These results underscore the importance of reforming the TFW management system in accordance with local conditions. In areas with abundant forest resources, the TFW's management system should shift to single leadership (jurisdictional or vertical management). In forest resource-scarce regions, the TFW's management system should change to dual leadership.

**Keywords:** township forestry workstation; collective forest; carbon density; management system; functions





## 1. Introduction

Protecting forests is crucial for maintaining ecological balance, ecosystem services, and biodiversity, while forest management is key to ensuring forest health and sustainable use. Approximately 48.33% of the world's nine billion hectares of forests and farmland are controlled by small-scale producers, local communities, and indigenous peoples [1]. Forests are a shared resource for communities that require the support and participation of residents and local governments to ensure sustainable management and protection. Additionally, community participation can yield social and economic benefits, such as employment opportunities, increased income for residents, and promotion of economic development [2–4]. Collaborative efforts involving various stakeholders, such as government and non-governmental organizations, have resulted in the widespread adoption of jointly formulated and implemented forest management plans [5,6].

The participation of government organizations in community forestry governance has been widely studied. On the one hand, government organizations empower communities with forest management rights by signing forest management agreements with them [7]. On the other hand, government participation can improve the responsibility and transparency of community forestry governance, and provide forestry research and education training to ensure the sustainability of forest protection [8]. In implementing forest fire and pest control projects, grassroots forestry institutions are responsible for forest risk management and providing technical and financial support to private forestry practitioners [9]. Currently,

government participation in community forestry still faces challenges such as distribution of benefits and fairness, lack of funds, and stakeholder conflicts [10,11]. However, China's unique "fragmented management" system also poses challenges to the functions and roles of grassroots forestry institutions [12]. This article focuses on the impact of China's forestry grassroots organizations and management system on community forestry, exploring the role of grassroots forestry organizations under different management systems, and providing direction for institutional reforms in the grassroots forestry sector in China and other developing countries.

Grassroots forestry institutions are widely distributed and numerous, which increases interaction with surrounding communities. Therefore, grassroots forestry institutions are familiar with the forestry situation of local communities and the needs of forest producers, and they can provide convenient, efficient, and accurate forestry management and service. Many countries have established grassroots forestry institutions to manage and protect the sustainable use of forest resources and indirectly increase forest carbon stocks [13–15]. An example is the Finnish Forest Management Association, which offers a variety of services based on the resources of individual associations and the needs of local forest owners [16]. The mission of the Forest Service in the United States is to develop science and technology extensively, serve the public, manage forest resources, maintain and sustainably use forest resources, collect information data needed for forest research, and provide technology and services for private forest owners [17]. The Japanese Forest Association is the final organization in charge of forest resource management, covering ten areas such as forest farmer guidance, forestry management, the forestry industry, and forest product processing. It plays a role in connecting forest farmers and the market [18]. The local forestry bureau in Germany is primarily in charge of ensuring that forests under various ownership comply with laws and regulations. In addition, it also helps private forest owners to formulate a series of plans for forestry management, cutting, and sales, make suggestions, and provide services [19].

The grassroots organization of forestry in China is the township forestry workstation (TFW). Since 1950, the development of China's TFW has gone through four important stages (Figure S1) [20–22]. In 2020, 11.71 million yuan was invested by the state and 149.8 million yuan by local governments in the construction of TFWs. At present, there are more than 22,000 TFWs in China. In 2015, the State Forestry Administration promulgated the Forestry Workstation Management Measures, which made it clear that TFW has six functions: policy publicity, resource management, administrative enforcement of law, production organization, science and technology promotion, and socialized services. In essence, the function of the TFW can be divided into two categories: a management function and a service function. Township forestry workstations can provide public management and services that small-scale private forest owners cannot produce. The management function is mainly to realize the government's administrative management of forestry and ensure the ecological function of forestry. The service function is to provide relevant services to foresters [23]. Qualitative research shows that TFWs, as the most basic management and service institutions of Chinese forestry, play an important support and guarantee role in China's collective forestry construction, forest resource protection, and collective forest tenure reform [11,21,24,25]. Until now, there has been no empirical study on the effect of TFWs on carbon sequestration in collective forest land.

The weakening of management functions and low serviceability of grassroots forestry institutions will not guarantee and guide private forest owners' production and management activities [26]. External factors such as personnel quality, institutional capital investment, infrastructure, and technical strength will affect the adequate performance of its functions. However, due to the unique administrative system of "piece partition" in China, the functions and roles of grassroots forestry institutions are also affected by different management systems [26]. Township forestry workstations have three management systems: jurisdictional management, vertical management, and dual leadership. Due to the unfavorable management system, a TFW has "big responsibilities, little rights, and weak

capabilities" and significant differences in functional performance [27]. China's TFW needs to improve due to its chaotic management system and unclear functional divisions [28]. In order to meet the needs of modern forestry development, TFW must carry out functional reform and organizational adjustment [29]. With the reform of withdrawing villages and combining towns in China, the TFW under the dual leadership system is the first to be withdrawn in large numbers. However, studies have explained that, compared with the single leadership management system, TFW under the dual leadership system is more likely to cause buck-passing in management and affect the efficiency of functional operations [27]. However, Liu et al. (2022) believed that the linkage mechanism between the two departments needs to be more cohesive under a single leadership in the TFW [28]. The above deficiencies could be avoided in the TFW under dual leadership. China has a vast landmass, and there are considerable differences in resource endowment and economic development levels among different regions, so the actual situation is relatively complicated. It is unrealistic for a TFW to adopt a unified national mode in the management system [27]. Therefore, it is necessary to change the functional awareness of TFWs completely, comprehensively consider the characteristics of regional resource endowment and the needs of forestry development, and reform the management system of TFW [30].

In summary, this article addresses the following issues: (1) Whether TFW in China improves the level of forest land carbon sequestration by small-scale forest farmers; (2) What the differences are in the main functions of TFW under different management systems. (3) How can TFW's management system be reformatted and adjusted in the future?

## 2. Materials and Methods

The scope of this study encompasses 31 provinces and cities in China, excluding Hong Kong, Macau, and Taiwan. China's provinces display considerable variation in terms of climate, geology, soil, and other conditions, which results in an uneven distribution of forest resources and significant differences in forestry development conditions. These disparities can impact forestry management and ecological restoration efforts, necessitating a consideration of regional differences and characteristics in order to develop more precise and effective policies and measures.

Over the past 70 years, China has established more than 22,000 forestry workstations in its towns and villages. These institutions serve as the fundamental comprehensive management service organizations within China's forestry management, constituting the endpoint of forestry work extension and a critical linkage between forestry work and forest farmers [12]. Forestry workstations form an essential component of China's forestry management, and while their number and scale vary across provinces, they perform an important role in the promotion of local forestry development and ecological environment protection. By studying the 31 provinces within our research area, we can explore the universality and applicability of forestry workstations on a larger scale, providing a more scientific basis and guidance for the construction and management of forestry workstations nationwide.

### 2.1. Data Collection

The data were obtained from the statistical yearbooks and databases of previous years. Each province's collective forest volume and forest area are from the 6th–9th National Forest Inventory Data. The functional variables (KAA, FTA, NFSH, TAA, TEA, HCD), independent variables (TFW, D_TFW, S_TFW, Staff), and control variables (IFP, FCR, FFI, GVF, AA) were obtained from the China Forestry and Grassland Statistical Yearbook, 2001–2021, AP from the China Environmental Statistical Yearbook, Temp from the Data Center for Resource and Environmental Sciences, Chinese Academy of Sciences (https://www.resdc.cn/, accessed on 25 August 2022). Instrumental variables (townships) obtained from the China Statistical Yearbook. After data matching and cleaning, this paper acquired relevant information from 31 provinces and cities from 2000 to 2020. The indicators FFI and GVF economic data were deflated, with 2000 as the base period.

## 2.2. Variable Measures

Collective forest carbon density. In this paper, "collective forest carbon density" refers to the amount of carbon sequestered by forest trees per unit of forest area at a particular time. The calculation of forest carbon density in the IPCC often includes three components:

- The amount of carbon sequestered by tree biomass;
- The amount of carbon sequestered by understory plants;
- The amount of carbon sequestered by forest land.

Due to the wide distribution of forest areas in China and the significant differences in natural climate between regions, in order to avoid introducing errors in the measurement of the amount of carbon sequestered, only the amount sequestered by tree biomass is included in the calculation of collective forest carbon density. It was found that tree biomass carbon sequestration accounted for 41% of total forest carbon sequestration, understory plant carbon sequestration accounted for 8%, and forest land carbon sequestration accounted for 51% [31]. The formula for calculating carbon density in collective forests is

$$C\_density_{ij} = V_{ij} + \delta + \rho + \gamma \tag{1}$$

In Equation (1), $C\_density_{ij}$ denotes the forest carbon density of forest type $j$ in type $i$; $V_{ij}$ denotes the unit storage volume of forest type $j$ in type $i$; $\delta$ is the biomass expansion factor and the default value of IPCC is 1.90; $\rho$ is the volume factor and the default value of IPCC is 0.5; $\gamma$ is the carbon content rate and the default value of IPCC is 0.5. Xi (2006) suggested that the above coefficients can be used for macroscopic estimation of forest carbon sequestration because they are the average values after individual factors are excluded [32]. However, there may be some errors if these coefficients are applied to a specific forest plot for carbon sink calculation.

Township forestry workstations. This paper uses each province's total number of TFW as the key independent variable. Due to the significant variation in forest land resources, different priorities of forestry tasks among Chinese provinces, and some regional policy changes, various management systems have been formed for TFWs in each province. The total number of TFWs can be calculated by aggregating the numbers of D_TFW and S_TFW.

Management system. A TFW has three management systems: jurisdictional management, vertical management, and dual leadership. Specifically, vertical management system refers to the unified vertical management of "people, finance, materials and affairs" of TFW by the county forestry bureau and the total allocation of personnel wages and funds from local finance, which guarantees the independence of township forest station in personnel and finance. Jurisdictional management means that the TFW is one of the departments of the township government, the personnel allocation is set and managed by the township government, and the level of capital investment and infrastructure construction is closely related to the local economic development level of the township. Dual leadership system means that the TFW is under the leadership of the township party committee and government and the operational guidance of the county forestry authorities. According to whether the TFW accepts independent decrees, management systems are divided into two types: single leadership system and dual leadership system. Combined with the main functions of TFWs, the paper explores the internal reasons for the differences in the impact of different management systems on the carbon density of collective forests.

Functions. According to the nine specific job responsibilities (R1–R9) in the Management Measures for Forestry Workstations issued by the National Forestry and Grassland Administration in 2016 (http://www.gov.cn/, accessed on 30 August 2022), as shown in Figure 1, six specific responsibility indicators were selected.

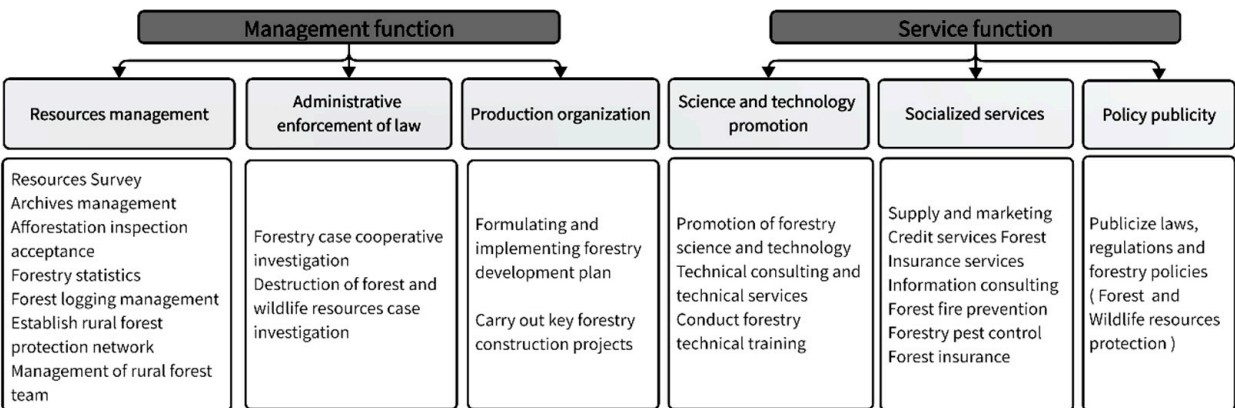

**Figure 1.** Main functions of township forestry workstations.

In the service function, the primary responsibility of the TFW policy publicity is to promote and implement the laws, regulations, various forestry guidelines, and policies on forest and wildlife resources protection (R1). In practice, TFW disseminates information to forest farmers through publicity vehicles, leaflets, and training, so the TFF (Training of Forest Farmers) indicator was selected for testing.

A TFW mainly establishes a science and technology demonstration base, providing technical training, consultation, and other services to help forest farmers solve the problems of low technology level and backward management modes. Socialized services refer to providing forest farmers with a series of public services, such as pest control, forest fire prevention, forest insurance, and mortgage financing of small forest rights, to help them better cope with natural and social risks and improve the quality of their forest land. In this paper, we chose TEA (Technology Extension Area) and NFSH (Number of Forestry Service Households) to measure the function of science and technology extension and socialization services in TFW, respectively (R8).

The resource management functions of TFWs include conducting resource surveys, carrying out forest harvesting and other administrative permit acceptance, establishing and improving village forest protection networks, and managing village forest protection teams (R3, R4, and R7). This paper uses the indicator FTA (Forest Tending Area) to measure the resource management and protection function of TFW.

Forest law enforcement mainly assists relevant departments in handling disputes over ownership or use of forests, trees, and forest land and investigating and dealing with cases of destruction of forest and wildlife resources (R6). This paper uses HCD (Handling Contract Disputes) to measure forest law enforcement.

The production and organizational functions of TFWs include assisting county forestry authorities and township people's governments in formulating and implementing forestry development plans, carrying out forest fire prevention, forestry pest control, forest insurance, and critical forestry construction projects (R2, R5). The research found that TFW supervises and inspects the situation of returning farmland to forest, natural forests, and key project reforestation monitoring. This paper uses KAA (Key Afforestation projects Area) to measure production and organizational functions. The statistical information of variables is shown in Table 1.

**Table 1.** Descriptive analysis of key variables (n = 651).

| Variable | Unit | Mean | SD | Variable | Unit | Mean | SD |
|---|---|---|---|---|---|---|---|
| | | Dependent variable | | | | Instrumental variable | |
| *C_density* | t/hm$^2$ | 17.01 | 11.12 | townships | 10$^3$ pcs | 1.111 | 0.802 |

**Table 1.** *Cont.*

| Variable | Unit | Mean | SD | Variable | Unit | Mean | SD |
|----------|------|------|-----|----------|------|------|-----|
| Function variable | | | | Control variable | | | |
| KAA | $10^4$ hm$^2$ | 7.421 | 9.086 | Natural environmental factor | | | |
| FTA | $10^4$ hm$^2$ | 19.58 | 23.45 | AP | mm | 1773 | 1449 |
| NFSH | $10^4$ HH | 9.829 | 10.97 | Temp. | °C | 12.58 | 6.199 |
| TAA | $10^4$ pt * | 26.43 | 36.66 | IFP | % | 7.333 | 7.33 |
| TEA | $10^4$ hm$^2$ | 5.869 | 19.27 | FCR | % | 29 | 17.82 |
| HCD | pcs | 498.8 | 1320 | Economic environmental factors | | | |
| Independent variable | | | | FFI | $10^6$ RMB | 1025 | 1447 |
| TFW | $10^3$ pcs | 0.91 | 0.573 | GVF | $10^6$ RMB | 3.25 | 1.881 |
| D_TFW | $10^3$ pcs | 0.266 | 0.307 | Social environmental factors | | | |
| S_TFW | $10^3$ pcs | 0.656 | 0.42 | AA | $10^6$ hm$^2$ | 1.95 | 1.762 |
| Staff | $10^3$ p | 4.055 | 3.006 | CFTR | - | 0.571 | 0.495 |

\* pt (person-time).

### 2.3. Method

In this model, individual fixed effects and time fixed effects are introduced to control all effects that remain constant across individuals and time, thereby eliminating the confounding effects of these factors. As a result, the TW_FE model can more accurately assess the impact of the independent variable on the dependent variable, while avoiding the influence of the confounding effects of time and individual factors on estimation results.

In this study, the TW_FE model was selected for empirical analysis to control for the effects of natural, economic, and social factors on collective forest carbon density. Control variables for natural, economic, and social environmental factors were introduced, and individual fixed effects and time fixed effects were used to control for unobserved individual and time factors, enabling a more accurate evaluation of the impact of TFW on collective forest carbon density. Therefore, the TW_FE model is a suitable panel data model for this study.

In order to accurately evaluate the effect of TFW on collective forest carbon density, relevant variables were collected from the three aspects of natural, economic, and social factors, respectively, for control. The control variables were set based on existing studies by considering the natural, economic, and social environmental factors. As shown in Table 1, natural environmental factors include forest cover rate (FCR) [33], annual precipitation (AP), temperature (Temp), and incidence of forest pests (IFP). Economic and environmental factors include forestry fixed investment (FFI) and the gross output value of forestry (GVF). Social environmental factors include afforestation areas (AA). The article sets the dummy variable CFTR to control the effect of the Collective Forest Tenure Reform on carbon density in collective forests in China, with CFTR = 1 if the time is greater than 2008 and CFTR = 0 otherwise. There is evidence that small-scale producers with land tenure rights tend to make long-term investments in their land and forests (e.g., improved forest management, afforestation, and soil and water management) compared to small-scale producers with no land tenure security or only short-term security [34–37]. However, there is still a possibility of missing variables (e.g., time-invariant, and individual-invariant). Therefore, the bidirectional fixed effect was chosen for empirical study. The specific model is shown in:

$$lnC\_density_{it} = \theta_0 + \theta_1 TFW_{it} + \theta_2 \sum Control_{it} + \tau_i + \vartheta_t + \varepsilon_{it} \tag{2}$$

In Equation (2), the explanatory variable $C\_density_{it}$ represents the carbon density of collective forest in the province $i$ in the period $t$; the explanatory variable $TFW_{it}$ represents the number of township forestry stations in province $i$ in period $t$. $\sum Control_{it}$ refers to the control variables of nature (AP, Temp, IFP, FCR), economy (FFI, GVF), and society (AA, CFTR), with variable definitions as above. $\tau_i$ and $\vartheta_t$ represent regional fixed effects and time fixed effects, respectively; $\varepsilon_{it}$ is the residual term. $\theta_0$ represents the constant coefficient, $\theta_1$ and $\theta_2$ represent the coefficients of the respective variables.

In the context of panel data models, a fixed effects approach can be employed to account for the effects of individual and time fixed effects. However, in the presence of endogeneity issues, such as measurement errors or omitted variables, the fixed effects model is inadequate for eliminating the endogeneity bias. Consequently, the use of exogenous variables as instrumental variables is necessary to address endogeneity problems.

This study focuses on regions abundant in forestry resources, where the establishment of more TFWs to meet forestry management needs may give rise to a bidirectional causal relationship between TFW and collective forest carbon density (According to Article 7 of the Administrative Measures for Forestry Workstations of the State Forestry Administration "Where there are tasks of forestry production and management, forestry stations shall be set up in townships; Forestry stations can be set up in two or more townships where the tasks of forestry production, operation and management are relatively light."). As such, exogenous variables are needed as instrumental variables to address the endogeneity bias. The study selected the number of townships in each province ($township_{it}$) as the instrumental variable for forest stations, and employs the 2SLS method for empirical testing. The 2SLS method uses instrumental variables to address endogeneity bias, generating an unbiased estimate for the endogenous variable, which can then be used to estimate the coefficients of the original model. This approach effectively mitigates the impact of endogeneity issues, leading to more accurate estimation results. Thus, the 2SLS method is a suitable choice.

First stage:

$$township_{it} = \theta_0 + \theta_1 TFW_{it} + \theta_2 \sum Control_{it} + \tau_i + \vartheta_t + \varepsilon_{it} \tag{3}$$

Second stage:

$$lnC\_density_{it} = \theta_0 + \theta_1 \hat{township}_{it} + \theta_2 \sum Control_{it} + \tau_i + \vartheta_t + \varepsilon_{it} \tag{4}$$

In Equation (3), the explained variable $\hat{township}_{it}$ refers to the fitted value of the number of townships; the endogenous variable $TFW_{it}$ represents the number of township forestry stations in province $i$ in period $t$. In Equation (4), the explanatory variable $C\_density_{it}$ represents the carbon density of collective forest in the province $i$ in the period $t$. In Equations (3) and (4), $\sum Control_{it}$ refers to the control variables of nature (AP, Temp, IFP, FCR), economy (FFI, GVF), and society (AA, CFTR), with variable definitions as above. $\tau_i$ and $\vartheta_t$ represent regional fixed effects and time fixed effects, respectively; $\varepsilon_{it}$ is the residual term; $\theta_0$ represents the constant coefficient, $\theta_1$ and $\theta_2$ represent the coefficients of the respective variables.

## 3. Results

### 3.1. Statistical Analysis

Figure 2 depicts the dynamic change process of China's TFWs and collective forest carbon density from 2000 to 2020. The number of TFWs decreased from 36.643 in 2000 to 22.220 in 2020. The decline in TFW is mainly due to the "removing villages and merging towns" reform by local governments in China [28]. The carbon density of collective forests showed a downward trend and then an upward trend. In 2020, the average carbon density of the stand layer of communal forests in China was 30.11 t/hm$^2$. Intuitively, there is a negative correlation between China's TFWs and the carbon density of collective forests. Does this mean that a TFW harms the carbon fixation level of communal forests? Therefore, further empirical tests were needed to explore the real impact of TFW on the carbon density of collective forests.

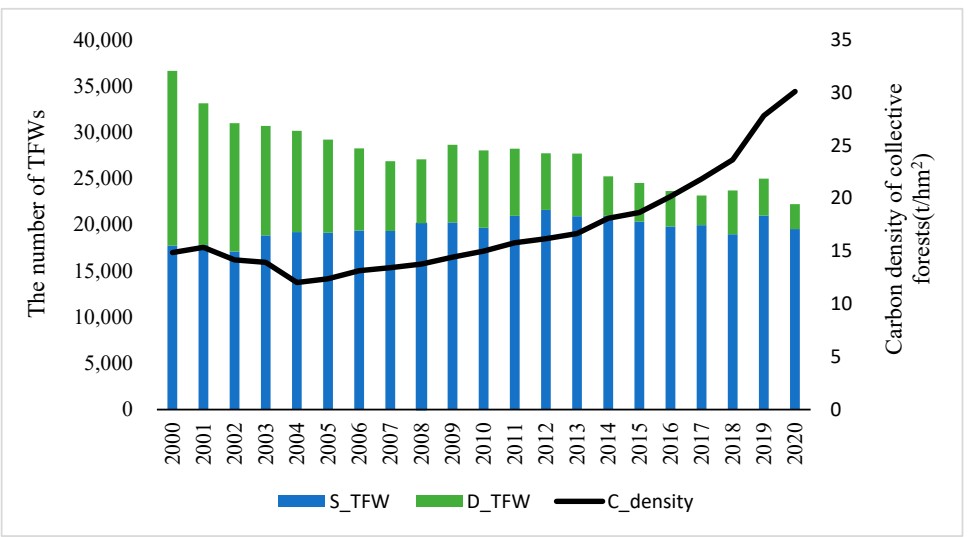

**Figure 2.** The evolution of TFW and collective forest carbon density from 2000 to 2020 (n = 651).

*3.2. Baseline Regression*

Table 2 gives the empirical results of OLS, Re, and FE_TW for the impact of TFWs on collective forest carbon density. From Column (3), the regression coefficient of TFWs is significantly positive at the 10% statistical level, indicating that a TFW positively affects collective forest carbon density; i.e., an increase of one unit ($10^3$) of TFWs increases collective forest carbon density by 9.1%, an increase of $100 \times (\exp 0.091 - 1) = 9.527$ t/hm$^2$. We used a random effects (Re) model as robustness tests for the attenuation bias caused by measurement error in the carbon density data (the attenuation bias of the random effects model is smaller if present), and the results of the random effects model are shown in Table 2 and are generally consistent with those of the fixed effects model. The significance and corresponding explanation of control variables are shown in the Table S1.

**Table 2.** Baseline regression results of the impact of TFW on carbon density of collective forests.

|  | (1) | (2) | (3) |
|---|---|---|---|
|  | **OLS** | **Re** | **Fe_TW** |
| TFW | 0.458 * | 0.108 ** | 0.091 * |
|  | −0.224 | −0.046 | −0.049 |
| Control | n | y | y |
| year | n | y | y |
| province | n | n | y |
| cons | 2.082 *** | 1.971 *** | 2.141 *** |
|  | −0.317 | −0.266 | −0.182 |
| N | 633 | 612 | 612 |
| r2 | 0.079 | - | 0.746 |

Standard errors in parentheses. * $p < 0.1$, ** $p < 0.05$, *** $p < 0.01$.

*3.3. Robustness Test*

Performing the panel without considering endogeneity in the fixed effects model estimation, which can lead to biased and inconsistent regression results. Since the collective forest carbon density data are truncated (non-negative), to avoid causing estimation bias, this study adopted a fixed effects Tobit model for re-estimation, and the empirical results are shown in Column (1) of Table 3. The regression coefficient of TFW is significantly positive at the 5% statistical level, indicating that TFWs positively impact collective forest carbon density; i.e., with an increase of one unit ($10^3$) of TFWs, the collective forest carbon density increases by 9.5%, and the results remain robust.

**Table 3.** Robustness test results of impacts of TFW on carbon density of collective forests.

|  | (1) | (2) | (3) | (4) | (5) | (6) |
|---|---|---|---|---|---|---|
|  | **Tobit** | **Fe_TW** | **Fe_TW** | **Fe_TW** | **Fe_TW** | **IV_2SLS** |
| TFW | 0.093 * | | | | | 0.230 *** |
|  | (0.048) | | | | | (0.078) |
| Staff | | 0.032 *** | | | | |
|  | | (0.009) | | | | |
| D_TFW | | | 0.125 ** | 0.132 * | | |
|  | | | (0.060) | (0.074) | | |
| S_TFW | | | 0.048 | | 0.006 | |
|  | | | (0.062) | | (0.076) | |
| townships | | | | | | 0.687 *** |
|  | | | | | | (0.125) |
| Control | y | y | y | y | y | y |
| year | y | y | y | y | y | y |
| province | y | y | y | y | y | y |
| _cons | 1.470 *** | 2.105 *** | 2.188 *** | 2.205 *** | 2.272 *** | 1.969 *** |
|  | (0.157) | (0.112) | (0.172) | (0.168) | (0.164) | (0.143) |
| N | 612.000 | 605.000 | 604.000 | 610.000 | 604.000 | 612.000 |
| r2 | | 0.759 | 0.756 | 0.755 | 0.751 | 0.737 |

Standard errors in parentheses. * $p < 0.1$, ** $p < 0.05$, *** $p < 0.01$.

Substitution of critical explanatory variables. As TFWs have been streamlined, staff have been continuously reduced. This article uses the number of staff on duty at the end of the year as a proxy variable for TFW, indicating its size. In Table 3, the regression coefficient of staff is significantly positive at the 1% statistical level, which means that each additional unit ($10^3$) of staff increases the collective forest carbon density by 3.2% on average. The average staffing of TFWs in 2020 (80,636/22,220 = 3.629) was less than four people which means that each additional unit ($10^3$) of TFW makes the collective carbon density increase by 11.6% on average, which is similar to the results in Table 2 (according to the staffing requirements of the Forestry Workstation Construction Standards for Townships, the staffing of primary stations in mountainous and mid-hill areas is 7–10, secondary stations are 5–7, and tertiary stations are 3–5, while primary stations in plain and pastoral areas are 5–7, secondary stations are 4–5, and tertiary stations are 3–4).

This study also used D_TFW and S_TFW to replace TFW, respectively. Column (4) indicates that the regression coefficient of D_TFW is significantly positive at the 10% statistical level, indicating that D_TFW has a positive effect on the collective forest carbon density; i.e., an increase of one unit ($10^3$) of D_TFW increases the collective forest carbon density by 13.2%. Column (5) indicates that the regression coefficient of S_TFW does not pass the 10% statistical level test, but the coefficient is positive, indicating that S_TFW positively affects collective forest carbon density.

Referring to the research of Liu (2009), the relational criticality test was used to test the effectiveness of different management systems [38]. According to the changes in variance that can be obtained, $\Delta R^2_{(3)-(4)} = \Delta R^2_{(3)} - \Delta R^2_{(4)} = 0.756 - 0.755 = 0.001$. $\Delta R^2_{(3)-(4)}$ represents the proportion of variance that S_TFW can account for collective forest carbon density, and $\Delta R^2_{(3)-(5)} = \Delta R^2_{(3)} - \Delta R^2_{(5)} = 0.756 - 0.751 = 0.005$. $\Delta R^2_{(3)-(5)}$ represents the proportion of variance that D_TFW can account for collective forest carbon density. As with $\Delta R^2_{(3)-(4)} < \Delta R^2_{(3)-(5)}$, D_TFW becomes more advantageous as collective forest carbon density rises.

### 3.4. Endogenous Test

This study chose IV-2SLS to solve the endogenous problem (two-way causality). According to the requirement of instrumental variable selection, the number of townships was chosen as the instrumental variable. On the one hand, the number of townships is an exogenous variable of the model, and the number of townships in each province remains

constant in the short term and has no direct causal relationship with collective forest carbon density. On the other hand, since 2004, China has carried out a comprehensive reform of township institutions and township abolition, and the institutional set-up of TFW has been dramatically affected by the abolition or merger of institutions [27]. Therefore, it is reasonable to choose the number of townships as the instrumental variable for TFW, but statistical tests are needed.

There are two main aspects of the instrumental variable test: the unidentifiable test and the weak instrumental variable test. The value of Kleibergen Paap rk LM statistic is 20.479, which rejects the hypothesis of unidentifiability at the 1% significant level, so the instrumental variables selected in this paper do not have an unidentifiability problem. Meanwhile, the Cragg Donald Wald F-statistic value is 128.657, which is significantly greater than the critical value of Stock Yogo's weak instrumental variable at the 10% significance level of 16.38. Therefore, the instrumental variables selected in this paper are not weak. In summary, the instrumental variables selected in this paper are very effective. As shown in Column (6), the baseline model underestimates the regression coefficients by the endogeneity problem, and the regression coefficients of TFW are significantly positive at the 1% statistical level, implying that each unit ($10^3$) increase in township forestry workstations makes the collective carbon intensity increase by 23.0% on average.

*3.5. Management System and Function of TFW*

The above research found that forestry workstations can improve the carbon density of collective forests, but this effect varies greatly due to different management systems. Liu et al. (2022) believe that there is a linkage mechanism between two departments for a dual leadership forest station, which has advantages in dealing with some complex forestry problems (forest right disputes), unified planning and layout of forestry (production organization), and formulating policy rules [28]. From the perspective of organizational economics and work design, the different nature of different tasks will have diverse impacts on the balance of the relationship between central government and local government, which required us to reasonably evaluate the nature of different tasks at present when choosing the management system [39].

Service function. The empirical results are shown in Table 4, where the regression coefficient of D_TFW(S_TFW) on TFF was found to be significantly positive at the 1% statistical level. In the policy propaganda function, D_TFW plays a better role than S_TFW, and its marginal effect is 2.210 times higher than that of S_TFW. Under different management systems, the technical extension and socialization service functions show significant differences. In the socialization service function, only the regression coefficient of S_TFW is significantly positive at the statistical level of 10%. In the forestry technical extension function, the marginal effect of S_TFW is better than that of D_TFW, and its marginal effect is 4.105 times that of D_TFW.

Management function. The empirical results show that the regression coefficients of D_TFW and S_TFW on the indicator KAA are significantly positive at the 5% and 10% statistical levels, respectively, indicating that this suggests a positive role for TFWs in the organization of forestry production. Meanwhile, the marginal effect of D_TFW is 1.367 times higher than that of S_TFW on forestry production organization. The regression coefficients of D_TFW and S_TFW on the indicator FTA are significantly positive at the 5% statistical level, indicating that TFWs positively affect forestry resource management. We also found that the marginal effect of S_TFW is better than D_TFW in a resource management function, but the difference between the two systems is not significant. It was found that the regression coefficient of D_TFW on the indicator HCD was significantly positive at the 5% statistical level, indicating that the more D_TFW there are, the stronger the forestry law enforcement capacity. The differences in forestry law enforcement functions under different management systems are significant, and the coefficient of S_TFW failed the test.

**Table 4.** Empirical results of functional differences in TFW under different management systems.

| | Service Function | | | Management Function | | |
| --- | --- | --- | --- | --- | --- | --- |
| | (1) | (2) | (3) | (4) | (5) | (6) |
| | TFF | TEA | NFSH | KAA | FTA | HCD |
| D_TFW | 75.510 *** | 3.065 ** | 4.887 | 9.469 ** | 11.056 ** | 389.910 ** |
| | −22.111 | −1.387 | −6.294 | −3.743 | −4.632 | −163.503 |
| S_TFW | 34.162 ** | 12.583 * | 8.656 * | 6.925 * | 11.373 ** | 353.503 |
| | −14.102 | −6.96 | −4.612 | −3.401 | −4.89 | −230.824 |
| Control | y | y | y | y | y | y |
| year | y | y | y | y | y | y |
| province | y | y | y | y | y | y |
| _cons | −49 | −5.034 | −65.485 | 3.132 | −14.579 | −322.212 |
| | −53.091 | −7.751 | −39.228 | −4.398 | −14.178 | −529.914 |
| N | 212 | 593 | 187 | 595 | 599 | 555 |
| r2 | 0.212 | 0.079 | 0.073 | 0.304 | 0.128 | 0.038 |

Standard errors in parentheses. * $p < 0.1$, ** $p < 0.05$, *** $p < 0.01$.

## 4. Discussion

We employed econometric models to investigate the contribution of forest stations in rural towns to the increase in collective forest carbon density using large sample data from 31 provinces and cities in China spanning the period from 2000 to 2020. Furthermore, we empirically analyzed the impact of forest station management systems on collective forest carbon density and explored the reasons for the differential effects of two management systems from the perspectives of organizational economics and job design. Our results demonstrate that: (1) China's rural town forest stations significantly enhance collective forest carbon density by fully exploiting their forestry management and service functions. (2) The impact of forest stations on collective forest carbon density varies significantly across different management systems, with D_TFW exerting a more substantial effect on collective forest carbon density. (3) In addition to FTA, the performance of the other five functions of the two management systems also differs significantly. D_TFW outperforms S_TFW in TFT, KAA, and HCD, while S_TFW performs better than D_TFW in TEA and NFSH functions.

In this study, we investigated the impact of government organization interventions on collective forestry carbon density in China. Our findings demonstrate that TFWs significantly increased the carbon density of collective forests. Other related studies have also revealed that grassroots government organizations play a crucial role in promoting the development of community forestry through policy and regulation formulation, funding and technology provision, and the establishment of management mechanisms [40]. Divya (2019) discussed the roles of government and non-government organizations in community forestry in the Indian Himalayas, and concluded that the integration of government and non-government organizations can generate synergistic effects [41]. Overall, it is evident that external organizations play an essential role in supporting community forestry resources. Forest resource protection and management are effective methods for forest carbon storage. Nevertheless, unlike previous qualitative studies, we estimated the degree of influence of grassroots forestry stations through econometric models.

Our study focused on the differences in the impact of government management systems on community forestry carbon density. The results indicate that D_TFW (dual leadership system) had a significantly higher impact on collective forest carbon density than S_TFW (single leadership system). The research findings pose a challenge to China's grassroots forestry institutional management system reform, as the number of D_TFW is declining year by year. Currently, the guidance for township forestry workstations is still based on uniform standards and unified functional requirements, without fully considering regional differences. Liu et al. (2022) believe that the linkage mechanism between the two departments of D_TFW has advantages in dealing with some complex forestry issues such as forest tenure disputes, unified planning and layout of forestry,

and formulation of policy rules [28]. However, some studies have shown that compared with a single leadership management system, a TFW under a dual leadership system is more likely to lead to management responsibility shifting and affect the efficiency of functional operations. In fact, we cannot draw a conclusion on which management system is better than the other since they perform significantly differently when facing different functional choices. D_TFW performs better than S_TFW in the TFT, KAA, and HCD functions, while S_TFW performs better than D_TFW in the TEA and NFSH functions. The difference in community forestry carbon density caused by management systems is partially due to functional differences. In studies on the roles and impact of government and non-government organizations in community forestry, Divya (2019) believes that the government's role is particularly important in policy and regulation formulation, while NGOs' roles are more prominent in the development and management of community forests [41].

The empirical results indicate that for each unit ($10^3$) increase in the number of township forestry workstations, the average carbon density of collective forests increases by 23.0% (see Table 3, Column (6)), which means that the carbon sequestration capacity of collective forests can be enhanced by increasing the number of TFWs. As shown in Figure 1, the production and organization function of TFWs includes assisting county forestry authorities and township governments in formulating and implementing forestry development plans. Township forestry workstations are responsible for supervising and inspecting afforestation monitoring of land returned from farming, natural forests, and key projects [27]. The law enforcement function of the forestry bureau mainly involves assisting relevant departments in handling ownership or use disputes over forests, trees, and forest land, as well as investigating and punishing cases of forest and wildlife resource destruction. The TFWs promote the flow of forest resources and expand the scale of forestry production by handling small-scale forestry and forestry contract disputes. The resource management and protection function of TFWs includes conducting resource surveys, obtaining administrative permits for tree felling, establishing and improving the rural forest protection network, and managing rural forest protection teams. Policy promotion mainly focuses on forest fire prevention, animal and plant protection, and prevention of forestry violations. The function of policy promotion can provide necessary policy information for forest farmers and enhance their legal awareness and green consciousness. Overall, the management and service functions of TFWs can directly or indirectly affect the growth of collective forest stock and improve the carbon sequestration level of collective forests.

The empirical results reveal that the impact of D_TFW on the carbon density of collective forests is significantly higher than that of S_TFW, suggesting that transforming the management system of TFWs can also enhance the carbon density of collective forests. One issue is that during the process of promoting forestry technology, TFWs' promotion initiative is often weak due to the low education level of forest farmers, resulting in poor actual results [26]. Furthermore, with the rapid increase in urbanization and the non-agricultural transfer of rural labor, some forest farmers no longer rely on forestry production as their primary household income source, which limits the function of social services. On the other hand, the noteworthy feature of D_TFW is its joint management by the county forestry bureau and the township government. The linkage mechanism between the two departments has advantages in dealing with some complex forestry issues (such as forest tenure disputes), developing unified forestry planning and layout (production organization), and establishing policy rules [28].

With the exception of FTA, there are significant differences in the performance of the other five functions between the two management systems, and this research conclusion provides direction for the reform of grassroots forestry department systems. In areas with abundant forest resources, the reliance on forest income has decreased, and changes in forest owners' attitudes toward their forest holdings have forced service providers to change their functions [42]. Foresters have, to some extent, taken on TFWs' role in forest management, such as afforestation and pest control. To better leverage TFWs' positive impact on collective

forest carbon density, they must complete the transformation from management functions to service functions. In areas with insufficient forest resources, decision-makers need unified forestry planning and policy guidance. In particular, afforestation, and reforestation must be based on local water resources. If planting depletes groundwater, exacerbating local water scarcity, negative impacts may result [43], necessitating TFWs' transformation from service functions to management functions.

This study contributes to the literature in two ways. First, unlike previous qualitative analyses, the positive impact of TFW on the carbon density of collective forests in China was examined through rich empirical analysis. This enriches the study of external organizational influence on community forestry and provides an important basis for the government to regulate public pool resources such as forests through institutional arrangements. Second, this study not only enriches the research on grassroots forestry institutions from a management system perspective but also demonstrates through empirical analysis how the government management system should be adjusted for better management and service of community forestry. This provides direction for the reform of grassroots forestry departments in China and other developing countries. However, our study also has limitations. Our data comes from national statistical data at the macro level, and we are unable to investigate the mechanism and impact of the functions of forestry workstations on the carbon density of collective forests.

## 5. Conclusions

China's commitment to achieving carbon neutrality by 2060 in response to the global challenge of climate change is laudable. Our research has uncovered the positive impact of forestry workstations in rural towns on the carbon density of collective forests. Therefore, it is imperative for rural towns in China to prioritize the construction of forestry workstations and scale up these workstations, particularly in regions with limited forest resources. The successful execution of policy initiatives such as the Three-North Shelter Forest System Project, Green Food Project, and Natural Forest Protection Plan hinges on the active participation of local governments, project funding, and technical support from the forestry department [44]. Consequently, more forestry workstations should be established in arid and semi-arid regions of China, to facilitate the implementation of large-scale ecological restoration projects and enhance the quality of forest land.

Our research findings indicate that the transformation of the management system of forestry workstations in rural towns has the potential to increase the carbon density of collective forests. Hence, policymakers should take into account the regional variations in the strategic guidance for TFWs' construction and adopt differentiated standards in terms of their setting form, management system, and personnel allocation, based on the available resources [28]. China's provinces differ significantly in terms of their climate, geology, soil, and other conditions, resulting in uneven distribution of forest resources and varying forestry development conditions. To meet the demands of local forestry production and management activities, TFW functions should be adjusted according to regional development needs. In general, we suggest that the adjustment of TFWs in each province should adhere to the principle of "local conditions." For areas with abundant forest resources, the management system of TFW should adopt a single leadership model (with jurisdiction and vertical management), whereas for areas with inadequate forest resources, a dual leadership model should be implemented.

While our research focuses on macro analysis, it is crucial to acknowledge the significant role of forest farmers as the main subject in the impact of grassroots forestry organizations on community forestry. Specifically, we must explore how the functions of township forestry workstations affect forest farmers at the micro level. To achieve this, we propose the collection of data through field investigations, which will facilitate a multidimensional analysis of the impact of forestry workstations on the production and management behaviors of small-scale forest farmers.

**Supplementary Materials:** The following supporting information can be downloaded at: https://www.mdpi.com/article/10.3390/f14030643/s1, Figure S1: The evolution of township forestry workstations; Table S1: Baseline regression results of the impact of TFW on the carbon density of collective forests. References [12,20–22,26,45–50] are cited in the supplementary materials.

**Author Contributions:** H.L. and S.L. contributed to the study's conception and design; data collection, and analysis were performed by L.G. and S.L.; the first draft of the manuscript was written by S.L. and K.H.S. All authors have read and agreed to the published version of the manuscript.

**Funding:** This work is supported by the key think tank project of Shaanxi Province on major theoretical and practical issues in philosophy and social sciences, "Ecological Product Value Realization Mechanism and Design of Shaanxi's Typical Ecological Product Value Realization Path" (2021ZD1041) and the Shaanxi Provincial Natural Science Project, "Research on the Foreign Exchange Increase Effect of Shaanxi State-owned Forest Farm Investment and Its Enhancement Strategy" (S2023-JC-YB-2373).

**Data Availability Statement:** No potential conflicts of interest are reported by the authors.

**Conflicts of Interest:** The authors declare no conflict of interest.

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
