# Peer review of "The Impact of Grassroots Forestry Institutions on Forest Carbon Sequestration: Evidence from China’s Collective Forests"

_forests, doi:10.3390/f14030643_

Round 1

Reviewer 1 Report

Dear authors,

It's a pleasure to go through the research and comment on possible improvement of the piece. I find the research interesting and novel. I will like to put forward few questions aimed at betterment of the MS.

1. Lines 43-46: how having large number and distribution can make grassroot forestry organisation has have natural management skills?

2. A discussion on importance of community participation on forestry, need for forest management through organizations needed in the introduction part before starting to focus on 'grassroot forestry'. This will give the readers idea on how importance is forestry followed by why grassroot forestry is required. Community participation in forestry through third sector is also gaining popularity-- https://www.sciencedirect.com/science/article/abs/pii/S1462901114001245, https://www.mdpi.com/2071-1050/15/1/368, https://www.sciencedirect.com/science/article/abs/pii/S0305750X18303486 etc

3. Fig- 2 and other tables/figures: number of data points need to be mentioned for better clarity (n=?)

Overall, I find the research relevant and novel. These minor comments are aimed at overall improvement of the piece and I will invite the authors to address the comments.

Author Response

Dear Reviewer,

I would like to express my deepest gratitude and respect to you. Your valuable time and effort in reviewing my paper are highly appreciated. Your careful reading and valuable suggestions and comments during the review process have been immensely helpful to me, and have allowed me to gain valuable experience and insights in writing my paper.

Your professional knowledge and experience have provided a more in-depth and comprehensive scrutiny and improvement of my research, making my paper more complete and excellent.

Once again, I want to express my heartfelt gratitude and appreciation for your support and encouragement towards my work.

Sincerely, Shuqiang Li

Reviewer 2 Report

There are some grammatical errors in the text.

-       Provide more justification for each mathematical definition and expression.

-       All notations used in equations should be defined.

-       The purpose of this research and the challenges related to the topic are not clearly stated in the introduction section.

-       The significance and background of selecting the study region need the inclusion of more information.

-       The clarity of figure 1 should be improved.

-       Discussion: The discussion section needs to be described scientifically. Kindly frame it along the following lines:

i. Main findings of the present study

ii. Comparison with other studies

iii. Implication and explanation of findings

iv. Strengths and limitations

v. Conclusion, recommendation, and future direction.

-       Please make sure your conclusions' section underscore the scientific value added to your paper, and/or the applicability of your findings/results, as indicated previously.

-       Please use the same format for all considered references.

On the whole, the manuscript contains rational elements, however, it requires major revision and should be redone significantly.

Author Response

(The authors gave the same response as above.)

Round 2

Reviewer 2 Report

The authors addressed satisfactorily all my concerns.

Author Response

Thank you again for your guidance, thank you very much